# The Battlefield of Chemotherapy in Pediatric Cancers

**DOI:** 10.3390/cancers15071963

**Published:** 2023-03-24

**Authors:** Letao Bo, Youyou Wang, Yidong Li, John N. D. Wurpel, Zoufang Huang, Zhe-Sheng Chen

**Affiliations:** 1Department of Pharmaceutical Sciences, College of Pharmacy and Health Sciences, St. John’s University, Queens, NY 11439, USA; 2Ganzhou Key Laboratory of Hematology, Department of Hematology, The First Affiliated Hospital of Gannan Medical University, Ganzhou 341000, China; 3Institute for Biotechnology, St. John’s University, Queens, NY 11439, USA

**Keywords:** chemotherapy, pediatric cancers, VIPN, DIC, MDR, drug transporters

## Abstract

**Simple Summary:**

The survival rate for pediatric cancers has improved significantly over the last decades. Conventional chemotherapies play a vital role in pediatric cancer treatment, especially in low- and middle-income countries, and the roster of chemo drugs for use in children has expanded. However, patients suffer from chemotherapy as a result of its countless side effects. Furthermore, multidrug resistance (MDR) continues to be an insurmountable obstacle that limits survival for a considerable number of patients. In this review, we discuss severe side effects in pediatric chemotherapies such as doxorubicin-induced cardiotoxicity (DIC) and vincristine-induced peripheral neuropathy (VIPN). Here, MDR mechanisms in chemotherapy are elucidated with the aim of improving survival, while also reducing the intensity and toxicity of chemotherapy. Furthermore, we focus on various drug transporters in common types of pediatric tumors, which could provide different therapeutic strategies.

**Abstract:**

The survival rate for pediatric cancers has remarkably improved in recent years. Conventional chemotherapy plays a crucial role in treating pediatric cancers, especially in low- and middle-income countries where access to advanced treatments may be limited. The Food and Drug Administration (FDA) approved chemotherapy drugs that can be used in children have expanded, but patients still face numerous side effects from the treatment. In addition, multidrug resistance (MDR) continues to pose a major challenge in improving the survival rates for a significant number of patients. This review focuses on the severe side effects of pediatric chemotherapy, including doxorubicin-induced cardiotoxicity (DIC) and vincristine-induced peripheral neuropathy (VIPN). We also delve into the mechanisms of MDR in chemotherapy to the improve survival and reduce the toxicity of treatment. Additionally, the review focuses on various drug transporters found in common types of pediatric tumors, which could offer different therapeutic options.

## 1. Introduction

Pediatric cancer is relatively rare and has a high survival rate compared with adult cancer. Pediatric cancer includes 12 major types and over 100 subtypes [1]. In the United States, pediatric cancer therapies have made remarkable advances over the last 70 years, and the overall survival (OS) rate for pediatric cancer patients has increased to 80% [2,3]. However, cancer is still the top cause of death by disease in children. In 2023, diagnosed pediatric cancer patients in the USA will include ~9910 children (birth to age 14) and ~5280 adolescents (aged 15–19 years). In addition, approximately 1040 children and 550 adolescents in the USA will die from cancer in 2023 [4]. An improvement in these amounts can be achieved by an increased understanding of the molecular basis of cancer through international collaborative efforts.

Despite these successes, the incidences and mortalities between countries are greatly different. In 2017, the 5-year survival rate of childhood cancer was 80% in high-income countries. However, data suggest that far fewer children survive in most low- and middle-income countries—the 5-year survival rate of children is ~40%, while more than 90% of children at risk of potential pediatric cancer live in these countries [5,6]. Because of major advances in modern science over the past several decades, the improvement in pediatric cancer in high-income countries has not translated to most low- and middle-income countries. There are still gaps among these countries regarding recognition, diagnosis, and treatment [7].

Despite financial hardship or treatment results, conventional chemotherapy is still a standard option for childhood cancer treatments in low- and middle-income countries, where molecular-targeted drugs are broadly accessible [8,9,10]. Severe side effects and resistance mechanisms against cancer therapies remain the primary reasons limiting chemotherapy outcomes [11]. Here, we consider the advancement of chemotherapy in pediatric cancer, including drugs and specific considerations. Beyond the previous focus on adult cancer alone, in the current review, we discuss severe side effects such as neuropathy and myopathy, the effects of the MDR phenomenon, and the role of various drug transporters in common types of pediatric tumors that could lead to more efficient pediatric chemotherapy. 

## 2. Current Chemotherapy in Pediatric Cancer

There are currently various cancer treatment approaches for cancer. The main four lines of treatment are surgical removal, immunotherapy, radiotherapy, and chemotherapy [12]. Surgical removal has the longest history in cancer treatment. It is an essential treatment option and often is often performed together with other cancer treatments, depending on the type of cancer [13]. Immunotherapy is generally considered to have been applied in 1890 by Dr. Coley, who injected a mixture of live and inactivated *Streptococcus* and *Serratia* to achieve responses [14]. It modulates the immune system to act better against cancer, and it includes immune checkpoint inhibitors, T-cell transfer therapy, monoclonal antibodies, and treatment vaccines. Radiotherapy precisely delivers radiation to kill cancer cells and shrink tumors, and was started in 1896 by Emile Grubbé, who treated a breast cancer patient using X-rays [15]. Chemotherapy was first introduced after the Second World War, and works by interfering with cell proliferation. Other cancer treatment approaches include stem cell transplantation, targeted therapy, photodynamic therapy, and hormone therapy.

The efficient treatment of pediatric cancer started after 1945. Sidney Farber treated a 3-year-old patient with acute lymphoblastic leukemia (ALL) with aminopterin [16]. As a conventional cancer therapy, chemotherapy has played an important role in pediatric cancer over the past several decades. In pediatric diagnosis, ALL is the most common type of cancer. The OS of ALL increased from 57% in the 1970s to 96% in recent years [17]. This enormous progress is partly attributed to advances in chemotherapy [3]. Currently, chemotherapy has reached its limit, as severe toxicities have been observed in pediatric patients during treatment. Modifications to chemotherapy are made in order to reduce multiple toxicities. For instance, studies have provided evidence that chronomodulated chemotherapy, based on the body’s intrinsic circadian clock, might minimize toxicity while maintaining an anticancer activity [18]. High-dose chemotherapy has been explored for treating childhood malignant glioma so as to pass the blood−brain barrier (BBB), reduce cell chemoresistance, and achieve a more comprehensive response [19].

Chemo drugs are grouped by function, chemical structure, and interaction with other medications. Types of chemo drugs include alkylating agents, antimetabolites, antibiotics, topoisomerase inhibitors, mitotic inhibitors, and corticosteroids [20,21]. In 1997, under the Food and Drug Administration Modernization Act2 (FDAMA), a pediatric exclusivity provision was enacted by the U.S. Congress. Few chemo drugs were explicitly approved for treating pediatric cancer. The FDAMA later reauthorized as the Best Pharmaceuticals Act for Children (BPCA), which encourages pediatric drug studies in companies by providing a financial incentive. A total of 16 chemo drugs have been approved by the FDA with pediatric indications in the U.S. Before the FDAMA, only nine chemo drugs were approved for pediatric indications, see Table 1. After the FDAMA (1997–2022), seven chemo drugs were approved to treat childhood cancer (see Table 2).

There are less drugs with pediatric indications worldwide compared with adult indications [22]. For some chemo drugs, the initial approval was only for adult indications. On 20 May 2022, Azacitidine was newly approved for childhood cancers, while it first approved on 19 May 2004 for treating all subtypes of myelodysplastic syndrome. Moreover, FDA approved these chemo drugs (Clofarabine, Nelarabine, Erwinia, and Mercaptopurine) for both adult and pediatric indications at the same time (Table 2). Although efforts have been made in pediatric chemotherapy, only a few chemo drugs have gained pediatric indications—there is a gap between pediatric and adult approval.

The therapy to treat pediatric cancers requires specific prerequisites and considerations, as patients are still growing and developing. Some adverse effects of cancer treatments are more severe for children than adults, as developing organs are more susceptible. Infection-related serious adverse events are more common in ALL patients receiving chemotherapy, which enhances the risk of death [23]. Therefore, it is necessary to identify drug resistance mechanisms to reduce chemotherapy intensity and toxicity. The counterstrategies against drug resistance are discussed in the following section.

## 3. Severe Neuropathy and Myopathy Side Effects in Chemotherapy

In cancer treatment, chemotherapeutic agents are considered a double edge sword, as it kills cancer cells and other healthy, fast-growing cells, without differentiation. As a matter of fact, patients suffer from various non-negligible side effects such as hair loss, fatigue, nausea, vomiting, and diarrhea. Currently, doxorubicin (DOX) and vincristine (VCR) are two commonly used chemotherapeutic agents in pediatric cancer treatment [25,26]. However, the potential harm of neuropathy and myopathy has become a considerate risk factor that limits the effect of DOX and VCR [27,28]. The pathway mechanism of neuropathy and myopathy induced by DOX and VCR in illustrated in Figure 1. Thus, it is necessary to discuss the opposing side and corresponding solutions for these two chemo drugs, mainly when applied to pediatric cancer patients.

DOX, an anthracycline drug to treat solid tumors in children, such as Wilm’s tumor, was approved by the FDA in 1974 to become the first clinical liposomal encapsulated anticancer drug [29]. DOX acts as an intercalation that inhibits topoisomerase II and further obstructs DNA replication [30]. The recommended dosage (2 mg/mL) is the same in both pediatric and adult patients. Symptoms of the side effects may include abnormality in electrocardiography, rhythm disturbances, and even left ventricular hypertrophy that leads to a reduction in ejection fraction [31].

DIC is a severe dose-dependent risk factor in cancer treatment [32]. It has been reported that SNP (rs2229774) in retinoic acid receptor-γ (RARG) has a significant impact on the increased occurrence of DIC. Further investigation found that a RARG agonist CD1530 showed a cardioprotective effect in an in vivo mouse model of DIC [33]. Other drug transporter genomic variants in the *SLC28A3* locus were also identified. Based on that, a single dosage of 3 µmol/L desipramine per day before the administration of DOX is recommended to prevent DIC [34]. Moreover, circular RNA (circRNA)-based therapy is a novel approach for treating DIC [35]. The overexpression of insulin receptor encoded circRNA (Circ-INSR) successfully prevented and reversed DIC in an in vivo mouse model [36]. Han. et al. reported another promising therapeutic target for DIC, namely tumor-suppressive human circular RNA CircITCH [37]. Other recent potential strategies that possibly reverse DIC include *atg7*-based autophagy activation [38], meteorin-like (METRNL) protein [39], sirtuin 1 (SIRT1) [40], berberine [41], ADAR2 [42], elabela (ELA) [43], phenylalanine-butyramide (FBA) [44], gasdermin D [45], melatonin [46], levosimendan [47], paeonol [48], SNX17 [49], irisin [50], isorhapontigenin [51], liensinine [52], etc.

VCR, known initially as Leurocristine, is the first-line chemotherapeutic medication often administered in the combination chemotherapeutic treatment of pediatric hematologic malignancies and solid tumors [53]. VCR acts as mitotic inhibitors by binding to the β-tubulin subunit of αβ-tubulin heterodimers, thus functionally destabilizing microtubule fibers, which ultimately leads to the termination of cancer cell division [54]. Previous studies have shown a cumulative effect for its neurotoxicity and overdosage may cause very serious or fatal outcomes [55,56]. Until 6 June 2022, it has been approved by FDA in the treatment of pediatric ALL, neuroblastoma, non-Hodgkin lymphoma, rhabdomyosarcoma, Wilm’s tumor, and other childhood kidney cancers. USP recommended dose of VCR injection for pediatric patients is 1.5–2 mg/m^2^ compared to adults is 1.4 mg/m^2^. For neonate and infant patients weighing 10 kg or less, the first dose should be 0.05 mg/kg, administered once a week. The latest research showed in the treatment of ALL and Wilm’s tumors, compared with older children, neonates or infants have similar clearance in vincristine. Thus, doses less than 0.05 mg/kg should not be applied in neonate and infant patients due to inappropriate suboptimal VCR exposures [57].

The common side effect encountered with VCR use is VIPN [58]. Clinical patterns of VIPN can be classified as sensory neuropathy (e.g., numbness and paresthesia), motor neuropathy (e.g., extremity weakness and walking difficulties), and autonomic neuropathy (e.g., constipation and urinary retention) [59]. VIPN can be assessed by the National Cancer Institute Common Terminology Criteria for Adverse Events (CTCAE), the pediatric-modified Total Neuropathy Scale (ped-m TNS), and the Total Neuropathy Score-Pediatric version (TNS-PV) [28,60]. There are currently no effective strategies for reducing vincristine-induced neurotoxicity. Also, whether VIPN exists chronically in the survivors remains ambiguous [61]. Some studies showed that a significant proportion of patients receiving VCR would undergo a certain extent VIPN [62,63]. In 2017, Tay et al. reported that ~16% of pediatric ALL survivors suffer from VIPN [64]. Nevertheless, in 2020, an investigation among 150 pediatric patients with ALL and Wilm’s tumors showed that significant side effects of the vincristine regimen are mostly neurotoxic, which is at a mild to moderate level [65]. In recent years, more and more studies have revealed the deep connection between genetic polymorphisms and VIPN [66,67,68]. The *CEP72* genetic variant is an optimistic VIPN signature marker since the *CEP72* gene encodes centrosome proteins that participate in the development of microtubules. Patients that carry *CEP72* TT genotype take potentially higher risk and severity of VIPN than CC or CT genotype patients [69]. Besides, the enzyme CYP3A5 contributes to hepatic clearance of VCR, which means CYP3A5 genetic polymorphisms, and its allelic variants are assumed to be associated with VCR neurotoxicity in different human populations. CYP3A5 genotyping analysis results showed over 70% of African Americans had been found to have one or more CYP3A5*1 alleles (such as CYP3A5 expresser), which is about five times higher compared to Caucasians [70]. Moreover, VCR is transported by some members of ATP-binding cassette (ABC) transporter superfamily such as ABCB1, ABCC1, ABCC2, and ABCB4 [71]. Lopez-Lopez et al. reported that ABCC1 is the critical mediator acts transporting VCR into the blood, while ABCB1 and ABCC2 are indispensable in the biliary excretion of VCR. The genotypes rs3740066 GG and rs12826 GG of ABCC2 were identified with significant associations with increased VIPN, which suggested that ABCC2 polymorphism could be used as a potential biomarker for VIPN in screening and diagnosis of pediatric ALL [72].

Recently, numerous updates have been related to the strategy to overcome VIPN. Zhou et al. reported levo-corydalmine ameliorates VIPN in mice by inhibiting Cx43 expression and NFκB-dependent CXCL1/CXCR2 signalling pathway [73,74]. Also, both VIPN and tumor growth were alleviated by the inhibition of histone deacetylase 6 (HDAC6) in mice. Other prospective targets that possibly prevent VIPN include mitoquinone [75], puerarin [76], nerve growth factor (NGF) monoclonal antibody DS002 [77], minocycline [78], bergapten [79], etc.

## 4. MDR: The Challenge in Pediatric Cancer Chemotherapy

Cancer cells can develop resistance to one chemotherapeutic drug, as well as other chemotherapeutic drugs that may have different chemical properties and mechanisms of action, which is called multi-drug resistance (MDR) [80]. MDR can occur in both adult and pediatric cancer chemotherapy. Although the mechanisms of MDR have been studied for a few decades, MDR is a very limiting factor to the success of cancer chemotherapies. MDR exists not only in chemotherapy and radiation therapy, but also in newly developed therapies such as targeted therapy and immunotherapy [81,82]. Drug resistance can be divided into intrinsic resistance or extrinsic resistance based on the cause of its occurrence. Cancer cells may have inherent drug resistance before receiving chemotherapy. However, various adaptive responses of cancer cells during the treatment cause extrinsic or acquired drug resistance of cancer cells. Several MDR mechanisms have been discovered, yet they are not yet fully understood. As shown in Figure 2, the factors of MDR include increased drug inactivation, reduction of influx and increased efflux of drugs, decreased activation of prodrugs, epigenetic dysregulation, changes in cell surface markers, tumor microenvironment (TME), epithelial−mesenchymal transition (EMT), altered miRNA, disruption of responses to DNA damage, inhibition of apoptotic pathways, tumor heterogeneity, and cancer stem cells [80,83,84].

## 5. MDR-Related Drug Transporters and Their Roles in Pediatric Cancers

Drug transporters are membrane proteins involved in the absorption, distribution, and excretion of drugs. Two transporter superfamilies have been identified in humans: the solute carrier (SLC) superfamily and the ABC superfamily [85]. ABC transporters are generally involved with the efflux of drugs, and SLC transporters have been chiefly described as influx transporters [86]. In addition to transporting therapeutic drugs across membranes, these transporters also mediate the transport of endogenous compounds. There is considerable interest in transporters from both families, as they are known to confer MDR to cancer cells.

### 5.1. SLC Family Transporters

SLC family transporters are a family that includes more than 300 membrane-bound proteins involved in the influx and efflux of a wide array of substrates, such as ions, metabolites, and drugs [87]. SLC transports substrates based on the electrochemical potential difference between the biological membrane or the ion gradient originally generated by the primary active transporters. Genetic variants of SLC transporters and clinical outcomes of methotrexate (MTX) have been studied in pediatric patients with ALL [88]. Although SLC transporter families are essential in human health, related studies focusing on pediatric cancer therapy are rare. We limited our discussion to ABC transporters.

### 5.2. ABC Transporters

The ABC transporter superfamily, one of the most prominent transporter families, is responsible for MDR by mediating drug efflux, which subsequently leads to a low intracellular concentration of antineoplastic agents in cancer cells and deteriorates therapeutic outcome [89]. In addition, more than efflux pumps, other critical roles in cancer development of this transporter superfamily have been revealed step by step [90,91,92,93,94]. So far, 7 subfamilies (ABC-A to ABC-G) and at least 48 additional subfamily members have been found and characterized depending on their structural differences and similarities [95,96]. Among these subfamily members, three members have been demonstrated that are closely related to MDR in chemotherapy, including P-glycoprotein (P-gp/ABCB1/MDR1), multidrug resistance protein 1 (MRP1/ABCC1), and breast cancer resistance protein (BCRP/ABCG2) [97,98].

#### 5.2.1. MDR1

The ABCB subfamily involves four full transporters (ABCB1/4/5/11) and 7 half transporters (ABCB2/3/6/7/8/9/10) that can transport a vast variety of molecules, including peptides, drugs, and ions [99]. Half transporters include two polypeptides, each having a transmembrane binding domain (TMD) and (nucleotide-binding domains) NBD to form a homo- or hetero-dimer. Full transporters are characterized as all four domains reside on a single polypeptide [97].

P-gp (also named ABCB1 or MDR1) is the first discovered and well-studied ABCB subfamily transporter that mediates MDR in cancer cells [100]. It is most expressed in the blood−brain barrier, liver, placenta, gallbladder, and endocrine tissues. In 2020, Nosol et al. revealed the protein structure of P-gp and found that inhibitors are bound in pairs and interact with structural features to block the function of P-gp [101]. As P-gp is the substrate of a broad range of antineoplastic agents, overexpressed P-gp has been demonstrated to lead to the development of drug resistance, including chemotherapeutic agents Vinca alkaloids (vinblastine and vincristine), Taxanes (paclitaxel and docetaxel), Anthracyclines (doxorubicin, daunorubicin, and epirubicin), and imatinib mesylate [102,103]. Meanwhile, a high-level expression of P-gp that causes cancer drug resistance has also been found in some tyrosine kinase inhibitors (Imatinib [104], GSK1070916 [105], and WYE354 [106]).

Altered expression levels of MDR1 have been detected in various tumors, such as neuroblastoma, rhabdomyosarcoma, and Wilm’s tumor. Neuroblastoma is the most common pediatric solid tumor, which accounts for 7–8% of childhood malignancies and 15% of all pediatric cancer deaths [107]. A study showed that the mRNA expression of MDR1 was increased in neuroblastoma patients with previous chemotherapy [108]. In addition, Qiu et al. [109] found that MDR1 hypermethylation expression can be associated with the pathogenesis and progression of neuroblastoma. MDR1 expression was observed to increase after chemotherapy in rhabdomyosarcoma and Wilm’s tumor [110]. These findings may be helpful to understand the role of MDR1 in pediatric malignancies regarding drug resistance and allow researchers to come up with strategies for therapeutic intervention.

#### 5.2.2. MRPs

Multidrug resistance proteins (MRPs) include 9 transporters from 13 members in the ABCC subfamily due to their ability to mediate cancer MDR [111].

Although each of the MRPs have slight differences in structures and amino acid compositions, the mechanism of transport driven by ATP hydrolysis is much the same. Unlike P-gp, which extrudes mostly xenobiotics, MRPs account for the extruding of both endo- and xenobiotics, thus showing its crucial role in regulating MDR processes in cancer development [112]. The structure of MRP1 has shown a novel substrate recruitment mechanism in that substrates are recruited straight from the cytoplasm, whereas P-gp attaches substrates from the inner leaflet of the lipid bilayer [113]. MRPs are distributed in the human body in various tissues, including the blood−brain barrier, brain, lung, kidney, liver, etc. [111].

The expression of MRPs has been investigated in several pediatric malignant tumors as they are a vital factor causing cytotoxic drug resistance and chemotherapy failure. Abnormal MDR expressions have been observed in pediatric malignancies, such as ALL, neuroblastoma, rhabdomyosarcoma, Wilm’s tumor, and retinoblastoma [114]. After chemotherapy, MRP1 expression has been observed to be upregulated in neuroblastoma, hepatoblastoma, and rhabdomyosarcoma patients [110]. Increased expression of MRP2-6 and decreased expression of MRP1 and MRP10 have been observed in ALL patients, which are associated with high doses of three chemotherapies [115]. Henderson et al. found that the inhibition of MPR1 is associated with reduced neuroblastoma development in transgenic mice [116]. These studies indicate their role in the chemotherapeutic drug efflux of MPRs and cancer prognosis.

#### 5.2.3. BCRP

The human BCRP has 665 amino acid residues with a molecular weight of 72 kDa. It is a half-transporter that is prominently expressed in various tissues, including, but not limited to, the brain, placenta, testis, liver, breast, and BBB [117]. A high overexpression of BCRP can be observed in different drug resistance cancer types, including in solid tumors and hematopoietic malignancies [118,119]. Although the clinical significance of BCRP-mediated drug resistance remains unclear, many studies have shown strong evidence to support that modulating the expression of BCRP could enhance drug sensitivity in chemotherapy [120,121].

BCRP was first identified in 1998, followed by expression studies to explore its potential role in chemoresistance [122]. The expression of the *BCRP* gene in childhood ALL has been observed at a low expression level [123]. However, compared with a diagnosis when the co-expression of BCRP and MDR1 was observed, a higher RNA level of BCRP was expressed at the relapsed/refractory state in acute myeloid leukemia (AML) [124]. Correlations between BCRP and MRPs have also been reported. The combined high expression of BCRP and MRP4 is correlated with reduced antileukemia drug methotrexate accumulation. Similarly, evaluated expression of the *BCRP* gene was found in primary neuroblastoma mitoxantrone-resistant cells [125]. These results underscore the potential value of BCRP as a predictor of chemoresistance drug efflux.

Essentially, MDR can arise because of altered targeted proteins or cell signaling pathways. Changes in cellular or non-cellular processes is also a significant factor of MDR. However, most encountered MDRs are related to drug efflux, which is mainly caused by ABC transporters. Once MDR transporters are overexpressed, the efflux of a chemo drug can increase. For example, the overexpression of P-gp, MRP1, and BCRP decreases the chemosensitivity of cancer cells by limiting exposure to anticancer drugs [126]. The overexpression of P-gp in cancer cells is associated with increased drug resistance to DOX and paclitaxel [127,128]. After anticancer agent treatment, the overexpression of P-gp has been found in acute myeloid leukemia [129]. Decreased chemosensitivity and a high expression of P-gp and BCRP were noticed in MDR patients with chronic lymphocytic leukemia, metastatic breast cancer, and multiple myeloma [130]. BCRP is regulated by proteins such as TGF-β1 and VEGFR-2 [131,132].

## 6. Tackling Strategies Regarding Drug Transporters

MDR is one of the barrier mechanisms against chemotherapy and thus is considered a key factor leading to the failure of chemotherapeutics. The overexpression of MDR1 that can enhance the efflux of cytotoxic agents is one of the targets to improve the effect of chemotherapeutics. One of the strategies is to reverse resistance mechanisms [133]. Several P MDR1 inhibitors have been investigated and shown successful MDR reversal on a drug-resistant prostate cancer cell line, without exhibiting a toxic potential [134]. Lei et al. [135] observed intracellular accumulation of paclitaxel and decreased drug efflux activity in the MDR1-knockout colorectal cancer cell line.

Furthermore, applying a nano-drug delivery system (NDDS) is a practical approach to enhance chemotherapy validity due to the targeted co-delivery, reduced sides affected, and long-time blood circulation achieved [136]. Curcumin has an antitumor activity and reverses the tumor MDR effect by regulating the MDR1 protein [137]. Degradable poly (lactic-co-glycolic acid) (PLGA) nanoparticles coloaded with curcumin and DOX could directly target cells or xenografted tumors and inhibit the growth of DOX-resistant esophageal carcinoma with a high biosafety [138]. NDDS-containing compounds have been developed that can inhibit MDR1, including tariquidar (XR9576), tetrandrine, verapamil, and cyclosporin A [139].

Therapy regimens can be different for treating children and adults as they have different drug-resistance profiles. The triple combination of fludarabine, ara-C, and G-CSF has been used in the treatment of childhood AML and caused an additive cell kill [140]. Tipifarnib can target the malignancies, such as leukemia, by activating RAS proteins (HRAS, KRAS, and NRAS) [141].

## 7. Special Considerations of Resistance in Pediatric Cancer Treatment

Pediatric malignancies have significant differences in their treatment compared with adult tumors, and thus require special considerations. Pan-cancer analyses have shown that pediatric cancers have comparatively lower mutation frequencies compared with adult cancers [142]. Epigenetic dysregulation, however, seems to be a particular factor in many types of pediatric cancers [143,144,145]. These genetic and non-genetic changes suggest therapeutic implications regarding chemotherapy.

Children with ALL have better prognoses and outcomes than adult patients with ALL. The 5-year OS rate is 87% for children aged 0–15 years, as opposed to 44% for adults aged 20–29 years [146]. A plethora of factors are responsible for the different outcomes, including socio-economic factors, resistance, disease heterogeneity, host responses, therapeutic treatment, etc. [147]. Resistance is one of the main factors leading to variables among ALL patients of different ages. For example, the activation of P-gp has a higher expression in adults [148]. In addition, the accumulated mutations in the *p53* gene and lower methotrexate polyglutamate may also contribute to the differences in the responses of drug resistance mechanisms [148,149]. Genetic lesions of polycomb repressor complex 2 (PRC2) have been reported in pediatric T-cell ALL (T-ALL), which promotes mutations of the IL7R/JAK/STAT pathway. Poor prednisone response and persistent MRD have been connected to adult T-ALL patients with loss-of-function alterations of PRC2 [150]. Genescà et al. reported that a complex karyotype (≥3 cytogenetic alterations) in adult T-ALL was associated with a minimal residual disease (MRD) level, but no correlation regarding the prednisone response [151]. Protein tyrosine phosphatase nonreceptor type 2 (PTPN2) is a phosphatase suppressing a gene in T-cell ALL. Deletions of PTPN2 in pediatric patients were associated with a higher glucocorticoid response and improved survival in children, yet these trends were not found in adults [149]. Other studies have suggested that a single subclone with additional mutations confers resistance to therapy, although half of the leukemia patients had multiple subclonal mutations [152]. Because children are not simply small adults, the difference in outcomes across age categories should be considered in diagnosis and therapy.

## 8. Clinical Trials and Recommendations for Risk Surveillance

With typically more aggressive protocols being used in children than in adults, chemotherapy is the standard treatment in cancers such as ALL, AML, and Hodgkin lymphoma [153]. The number of clinical trials related to chemotherapy-induced side effects is on the rise.

Notably, VIPN is being studied in two clinical trials. A Phase 4 clinical trial (NCT02923388) is testing Vitamin B12 and vitamin B6 in ALL patients treated with VCR.

The results from a study of 102 patients showed that vitamin B6 and B12 significantly reduced the incidence, relative risk, and severity of VIPN. The amount needed to treat was encouragingly low, and vitamin B6 and B12 were recommended as promising neuroprotective agents against VIPN [154]. A clinical trial (NCT02796365) is currently underway to evaluate the effectiveness of exercise rehabilitation as a preventive measure to DIC. The studies are summarized in Table 3.

Significantly, four clinical trials are investigating or studying DIC. A Phase 2 clinical trial (NCT04166253) is testing vitamin D in breast cancer patients treated with DOX. A clinical trial (NCT02796365) is currently underway to evaluate the effectiveness of exercise rehabilitation as a preventive measure for DIC. The studies are summarized in Table 3.

PanCare is Pan-European Network for care of pediatric cancer survivors. The risk for significant and potentially life-threatening late effects can be identified by certain long-term follow-up projects such as PanCareSurPass and PanCareFollowUp. Based on the reports from 10 countries, ototoxicity following platinum-based chemotherapy has been evaluated regarding the quality of evidence. The ototoxicity surveillance recommendations for pediatric cancer survivors’ future studies should focus on the evaluation of otoprotectants and the identification of optimal threshold doses to prevent ototoxicity [155]. In addition, candidate genetic markers are useful for identifying childhood cancer patients at risk of severe late effects, such as *SLC22A2,* which detects those at risk of platinum-induced hearing loss [156]. Moreover, adequate knowledge of cancer history, subsequent treatment exposure, and potential risks of late effects are needed to enhance survivors’ health and self-management skills. Accessible and reliable information is essential to increase awareness about late effects, which is necessary for providing personal recommendations for surveillance and prevention [157].

## 9. Conclusions

Together with the significant advancements in target therapy and immunotherapy, chemotherapy remains the primary treatment option for childhood cancer patients in most low- and middle-income countries. Recognizing the absolute validity of chemo drugs is crucial as they have shown to be highly effective for obtaining long-term survival in childhood cancers. Until now, in total, 16 chemo drugs have been approved by the FDA for pediatric chemotherapy. Unlike adult cancer patients, children diagnosed with cancer are less tolerant of chemotherapy. The numerous adverse effects and occurrence of MDR have become a significant obstacle in pediatric chemotherapy. It is crucial to alleviate severe side effects such as VIPN and DIC. Meanwhile, a comprehensive understanding of MDR and its reversal mechanism is essential. To improve follow-up care quality, the electronic document summarizes the clinical history of childhood/adolescent cancer survivors, including treatments received, and provides personalized follow-up and screening recommendations. By overcoming the challenges of traditional chemotherapy, we will hopefully be able to improve both the survival rate and overall quality of life of pediatric cancer patients.

## Figures and Tables

**Figure 1 cancers-15-01963-f001:**
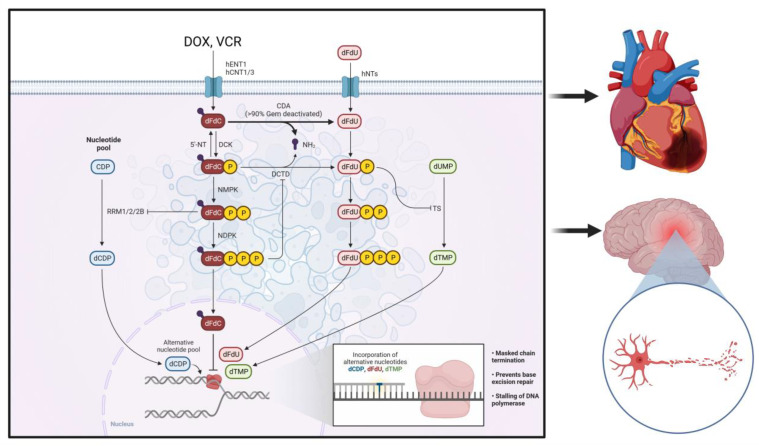
Neuropathy and myopathy induced by two commonly used chemotherapeutic agents (DOX and VCR) in pediatric cancer treatment.

**Figure 2 cancers-15-01963-f002:**
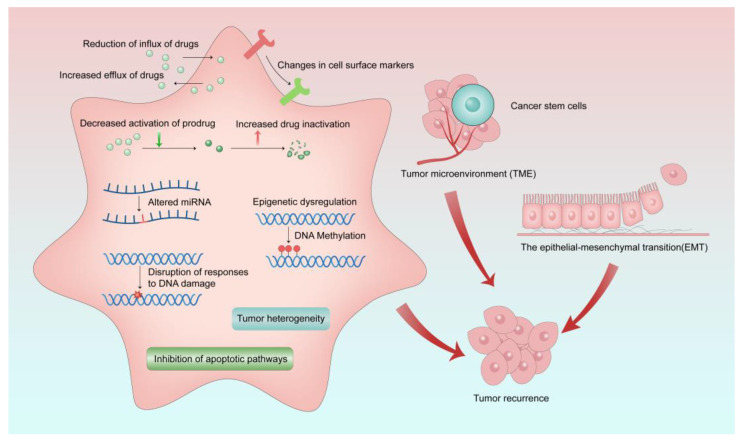
Mechanisms of MDR in cancer treatment. MDR occurs as a result of various causes, which ultimately lead to tumor recurrence.

**Table 1 cancers-15-01963-t001:** Chemo drugs approved prior to FDAMA, for which labeling includes pediatric indications [24].

Drug	Original Approval	Indications
Doxorubicin Hydrochloride	7 August 1974	Wilm’s Tumor and Other Childhood Kidney Cancers
Vincristine Sulfate	10 July 1963	ALL, Neuroblastoma, Non-Hodgkin Lyphoma, Rhabdomyosarcoma, Wilm’s tumor and other childhood kidney cancers
Cytarabine	17 June 1969	Acute Nonlymphocytic Leukemia
Cyclophosphamide	16 November 1959	ALL
Methotrexate Sodium (Trexall)	10 August 1959	ALL
Mercaptopurine (Purinethol, Purixan)	11 September 1953	ALL
Daunorubicin Hydrochloride (Rubidomycin)	19 December 1979	ALL
Procarbazine Hydrochloride (Matulane)	22 July 1969	Hodgkin Lymphoma
Dactinomycin (Cosmegen)	10 December 1964	Ewing sarcoma, gestational trophoblastic disease

Data provided by National Cancer Institute (https://www.cancer.gov/), accessed on 30 January 2023, updated: 20 December 2022, and U.S. FDA (https://www.fda.gov/). The drugs and drug combinations are not listed here.

**Table 2 cancers-15-01963-t002:** Chemo drugs approved post FDAMA with pediatric specific indications (1997–2022) [24].

Drugs	Original Approval	Pediatric Approval	Indications for Pediatric Cancer
(Drugs approved post FDAMA with pediatric specific indications (1997–2022))			
Azacitidine (Vidaza)	19 May 2004	20 May 2022	JMML
Calaspargase Pegol-mknl (Asparlas)	same	20 December 2018	ALL
Everolimus	1 November 2010	29 August 2012	Giant Cell Astrocytoma
Asparaginase Erwinia Chrysanthemi (Erwinaze)	same	18 November 2011	ALL
Clofarabine (Clolar)	same	28 December 2004	ALL
Pegaspargase (Oncaspar)	same	24 July 2006	ALL
Nelarabine (Arranon)	same	28 October 2005	Non-Hodgkin Lymphoma

Data provided by National Cancer Institute (https://www.cancer.gov/), updated: 20 December 2022, and U.S. FDA (https://www.fda.gov/). JMML: juvenile myelomonocytic leukemia. The drugs and drug combinations are not listed here.

**Table 3 cancers-15-01963-t003:** Clinical trials of DIC and VIPN.

Toxicity	Study Title	NCT Identifier	Phase	Patient Number	Disease	Status	Treatment/Method
DIC *	Protective Role of Vitamin D in Breast Cancer Patients Treated with Doxorubicin	NCT04166253	Phase 2	100	Breast cancer	Completed	Vitamin D
DIC	99mTc-rhAnnexin V-128 Imaging and Cardiotoxicity in Patients with Early Breast Cancer	NCT02677714	Phase 2	14	Breast cancer	Terminated	Radiation: 99mTc-rhAnnexin V-128
DIC	Prevention Using Exercise Rehabilitation to Offset Cardiac Toxicities Induced Via Chemotherapy (HF-PROACTIVE)	NCT02796365	Not Applicable	29	Breast cancer, Gastric cancer, Leukemia	Completed	Exercise
DIC	Evaluation of Myocardial Injury After Anthracycline Chemotherapy in Osteosarcoma Patients Using CMR	NCT04461223	Not Applicable	55	Osteosarcoma Myocardial Injury	Unknown	Contrast-enhanced cardiac magnetic resonance imaging, observational Study
VIPN ^#^	Neuroprotective Effect of Vitamin B12 and Vitamin B6 Against Vincristine Induced Peripheral Neuropathy	NCT02923388	Phase 4	88	Acute Lymphoblastic Leukemia (ALL)	Completed	Vitamin B12 and vitamin B6
VIPN	Physiologic Measure of VIPN	NCT04786977	Not Applicable	40	Chemotherapy-induced Peripheral Neuropathy	Recruiting	No Intervention, observational Study

Data provided by Clinical Trials (https://clinicaltrials.gov). * DIC-doxorubicin-induced cardiotoxicity; # Vincristine Induced Peripheral Neuropathy.

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
