# Peer review of "The Battlefield of Chemotherapy in Pediatric Cancers"

_cancers, 2023, doi:10.3390/cancers15071963_

Round 1

Reviewer 1 Report

The authors present a paper : " The Battlefield of Chemotherapy in Pediatric Cancers" very interesting and very well documented from a clinical and biological point of view.

I have only minor suggestions to improve the validity of the paper:

1. The Conclusions are too hasty. The Authors describe the negative effect of some chemotherapies and the MRD limit in the efficacy of these drugs.We have to remember the absolute validity of these drugs in obtaining long term survival in more than 80% of childhood cancers. I understand the aim to decrease the long term side effects of these drugs but we have also to balance the benefit of these treatments. Tpo this purpose in the References I didn't see any paper from PANCARE international  group that recently published important papers on the long term toxicity (e.g. cardiomiopathy or second tumors).

2. Please give based on this study some practical suggestion or workable modality. Conclusions are too generic and not reporting any "take home message". I know it is not easy but could be important for clinicians for their management of oncological therapies.

Reviewer 2 Report

The authors report the battlefield of chemotherapy in pediatric cancers.

1.     In the side effects in chemotherapy part, the authors should add the Summary Figure. It will be benefit for the reader.

2.     The authors should provide the clinical trial information and describe in the text.

Round 2

Reviewer 2 Report

none